# Molecular Dynamic Analysis of Carbapenem-Resistant *Klebsiella pneumonia*’s Porin Proteins with Beta Lactam Antibiotics and Zinc Oxide Nanoparticles

**DOI:** 10.3390/molecules28062510

**Published:** 2023-03-09

**Authors:** Rasha Elsayim, Abeer S. Aloufi, Yosra Modafer, Wafa Ali Eltayb, Alaa Alnoor Alameen, Samah Awad Abdurahim

**Affiliations:** 1Department of Botany and Microbiology, College of Science, King Saud University, Riyadh 11451, Saudi Arabia; 2Department of Biology, College of Science, Princess Nourah bint Abdulrahman University, P.O. Box 84428, Riyadh 11671, Saudi Arabia; 3Department of Biology, College of Science, Jazan University, Jazan 45142, Saudi Arabia; 4Biotechnology Department, Faculty of Science and Technology, Shendi University, Shendi 11111, Nher Anile, Sudan; 5Department of Pharmacology and Toxicology, College of Pharmacy, King Saud University, Riyadh 11451, Saudi Arabia; 6Department of Microbiology, College of Medicine, Al-Rayan Colleges, AL Madinah Al Monawara 41411, Saudi Arabia

**Keywords:** *Klebsiella pneumoniae* carbapenemase, zinc oxide nanoparticles, molecular dynamics, green synthesis, multidrug-resistant

## Abstract

To prevent the rapidly increasing prevalence of bacterial resistance, it is crucial to discover new antibacterial agents. The emergence of *Klebsiella pneumoniae* carbapenemase (KPC)-producing Enterobacteriaceae has been associated with a higher mortality rate in gulf union countries and worldwide. Compared to physical and chemical approaches, green zinc oxide nanoparticle (ZnO-NP) synthesis is thought to be significantly safer and more ecofriendly. The present study used molecular dynamics (MD) to examine how ZnO-NPs interact with porin protein (GLO21), a target of β-lactam antibiotics, and then tested this interaction in vitro by determining the zone of inhibition (IZ), minimum inhibitory concentration (MIC), and minimum bactericidal concentration (MBC), as well as the alteration of KPC’s cell surface. The nanoparticles produced were characterized by UV-Vis spectroscopy, zetasizer, Fourier-transform infrared spectroscopy (FTIR), and scanning electron microscopy (SEM). In silico investigation was conducted using a variety of computational techniques, including Autodock Vina for protein and ligand docking and Desmond for MD simulation. The candidate ligands that interact with the GLO21 protein were biosynthesized ZnO-NPs, meropenem, imipenem, and cefepime. Analysis of MD revealed that the ZnO-NPs had the highest log P value (−9.1 kcal/mol), which indicates higher permeability through the bacterial surface, followed by cefepime (−7.9 kcal/mol), meropenem (−7.5 kcal/mol), and imipenem (−6.4 kcal/mol). All tested compounds and ZnO-NPs possess similar binding sites of porin proteins. An MD simulation study showed a stable system for ZnO-NPs and cefepime, as confirmed by RMSD and RMSF values during 100 ns trajectories. The test compounds were further inspected for their intersection with porin in terms of hydrophobic, hydrogen, and ionic levels. In addition, the stability of these bonds were measured by observing the protein–ligand contact within 100 ns trajectories. ZnO-NPs showed promising results for fighting KPC, represented in MIC (0.2 mg/mL), MBC (0.5 mg/mL), and ZI (24 mm diameter). To draw the conclusion that ZnO-NP is a potent antibacterial agent and in order to identify potent antibacterial drugs that do not harm human cells, further in vivo studies are required.

## 1. Introduction

Due to the rapid spread of bacteria that produce extended-spectrum beta-lactamases (ESBL), carbapenems are being used more frequently as antibiotics of last resort to treat infections brought on by these multidrug-resistant (MDR) germs [1]. The most common species of these bacteria is *Klebsiella pneumoniae*, which is an antibiotic-resistant bacterium belonging to Enterobacteriaceae, responsible for many healthcare-associated outbreaks [2] The first *Klebsiella pneumoniae* carbapenemases (KPCs) were discovered in the USA in 1996. Since that time, these adaptable lactamases have proliferated worldwide among Gram-negative bacteria, particularly *K. pneumoniae*, even if their precise epidemiology varies across nations and regions [3]. *K. pneumoniae* resistance to meropenem and imipenem increased dramatically, rising from 2.9 and 3.0% in 2005 to 26.3 and 25%, respectively, in 2018. The Mediterranean and Balkan countries in Europe have the highest incidence of carbapenem-resistant *K. pneumoniae*, with rates of 60% in Greece and 40% in Italy [4] The Gulf Cooperation Council (GCC) countries are an excellent example of how international travel could pose a serious problem: a large number of their citizens travel to the United States and Europe for specialized medical care, a considerable portion of the population is made up of migrant workers from the Indian subcontinent, and millions of people travel to the region every year for the Hajj and other religious celebrations [5,6,7] In Saudi Arabia, concerns about carbapenem resistance among Enterobacteriaceae species are on the increase. Riyadh, in the country’s center, was where the majority of CRE-related reports occurred [8]. In a different recent study, the molecular characteristics of 54 isolates of *K. pneumoniae* that were carbapenem-resistant were isolated from clinical samples in 2 of the largest hospitals in Saudi Arabia’s southern province [9]. When bacteria acquire resistant genes encoding metallo-beta lactamases, carbapenem resistance can emerge. According to carbapenem hydrolysis and geographic distribution, the main carbapenem enzyme gens are OXA-48, NDM, KPC, VIM, and IM [2,5,9]. Gram-negative bacteria are exposed to carbapenems via outer membrane proteins (OMPs), referred to as porins. The Penicillin-binding proteins (PBPs) are “permanently” acylated by carbapenems after they have crossed the periplasmic gap. PBPs are enzymes that catalyze the synthesis of peptidoglycan in the cell wall of bacteria. Then, the peptidoglycan are weakened and the osmotic pressure eventually causes cell bursts [10]. Most KPC cases are treated with meropenem, imipenem, doripenem, and colistin. However, in recent years, the majority of KPC cases have developed resistance to all the previous antibiotics, including colistin. Based on the previous literature, it is clear that novel antibiotics are urgently needed. Recently, there has been extensive research conducted on the biological synthesis of nanoparticles to overcome resistant microbes. Zinc oxide nanoparticles are one of the most potent nanoparticles, reported by Elsayim and others as promising antimicrobial compounds [11,12,13]. In this research, we intended to evaluate the activity of biosynthesized ZnONPs against KPC in vitro and in silico. No prior study has examined the activity of zinc oxide nanoparticles against KPC in silico. The aim of the present study is to investigate the binding of ZnO-NPs, meropenem, imipenem, and cefepime to porin protein; showing the mechanism of the interaction by using molecular docking analysis; studying the stability of binding through molecular dynamic (MD) simulation analysis; and evaluating the activity of ZnO-NPs, meropenem, imipenem, and cefepime in vitro by using zone of inhibition, minimum inhibitory concentration, and minimum bactericidal concentration; and studying the surface alteration of the morphological bacteria’s cell.

## 2. Results and Discussion

As evidenced by numerous reports, beta lactam antibiotic resistance is dramatically rising. This calls for the urgent discovery of new ecofriendly antibiotics. In this project, we examined the interactions between ZnO-NPs, meropenem, imipenem, cefepime and KPC’s porin protein, demonstrating the interaction mechanism through molecular docking analysis, examining the stability of the interaction via simulation analysis, and testing the antibacterial activity of the previously tested antibiotics and ZnO-NPs in vitro through MIC and MBC. We used SEM to examine bacterial cell alterations before and after exposure to the tested antibiotics and Zno-NPs.

### 2.1. Zinc Oxide Nanoparticles’ Characterizations

Several synthetic compounds have antibacterial properties, but only a small number of them can be applied as biocides to create medications or coatings. Some of those compounds cannot be recommended as antibiotics because they are harmful to eukaryotic cells. ZnO and nano-ZnO compounds stand out among these materials as effective antibacterial agents and have been approved by the US’ Food and Drug Administration (FDA) as anticancer agents and for the treatment of various skin conditions [14,15]. In order to study ZnO-NP as an antimicrobial agent, we synthesized it by the green synthesis method, using *Acacia nilotica* fruits as a reducing agent [6]. Figure 1A–D shows the characteristics of the ZnO-NPs produced. UV-Vis absorbance spectrophotometry (wave range 200–800) was performed to confirm the formation of ZnO-NPs. ZnO-NPs’ absorbance spectrum revealed a significant absorbance peak at 367.5 nm (Figure 1A). This finding is in agreement with previous research findings that zinc oxide nanoparticle formation occurs at sizes between 300 and 390 nm [11,16,17]. The mean Z-average diameter (nm) was used as a preliminary screening to evaluate the size of ZnO-NPs. The size distribution was measured as 95.73 nm, which is considered a good result regarding size, in terms of antimicrobial compounds. This finding is consistent with that of Elsayim and others, who found that the size of ZnO-NPs was 94 nm [6]. To evaluate the potential of the synthesized nanoparticles, we used a zeta potential test; ZnO-NPs presented a strong anionic charge (−33 mV). This is considered a good result based on Clogston and Patri, who reported that strongly cationic and strongly anionic nanoparticles are those with zeta potentials of greater than +30 mV or less than −30 mV, respectively [18]. Another important finding is that studying the functional groups which are supposed to be responsible for the potential activity of the synthesized ZnO-NPs was detected by Fourier-transform infrared spectroscopy (FTIR) in the range of 400–4000 cm^−1^. In Figure 1C, ZnO-NPs display a sharp peak at 3429 cm^−1^, corresponding to the O-H-strong group. The group C-H medium and C-C group were shown in both the plant extract and ZnO-NPs while the N-H, C=O, and C-N groups were found only in *Acacia nilotica* extract, explaining the reducing attribute of the plant extract and demonstrating which functional groups mediated the formation of ZnO-NPs [6,19]. On the other hand, ZnO-NPs demonstrated two functional groups (-C-C-H, C-Br, and Z-O) at the peak’s end, indicating that zinc contributes to the production of ZnO-NPs. The final characterization test of ZnO-NPs was the study of the synthesized nanoparticles’ surface, size, shape, structure, and formation. Figure 1D illustrates the quaternary shape and aggregation of ZnO-NPs. Most of these particles had a smooth surface and appeared to be free of fractures. These results reflect those of Elsayim and others, who also found that their synthesized ZnO-NPs had aggregation features and a quaternary shape [11]. Although Elsayim and others synthesized their nanoparticles by *Acacia nilotica*, their nanoparticles presented a hexagonal shape, an outcome that was contrary to our result [6].

### 2.2. In Silico Studies

The docking scores of all bioactive components in the zinc oxide Versace carbapenem antibiotics meropenem, imipenem, cefepime, and KPC’s porin protein were assessed in the first round of analysis.

#### 2.2.1. Quantitative Structure–Activity Relationship (QSAR) Studies

To estimate the reactivity, potential, and characteristics of ZnO-NPs, meropenem, imipenem, and cefepime, QSAR studies were performed [20,21].

Examining Table 1, it is apparent that ZnO-NPs are more likely to be soluble in both organic and aqueous solvents, in comparison with other tested compounds, based on log P value. On other hand, all tested compounds displayed good stability, since they had low total energy [20,21]. In addition, they had acceptable QSAR properties, represented as solvent-accessible surface area and volume, and high surface area, molar refractivity, and polarizability, with the best results displayed by ZnO-NPs.

#### 2.2.2. Molecular Docking Analysis

##### Interaction Analysis of Protein

As shown in Figure 2, the 3D docking program detected that all ligands interacted with porin protein by docking in the same binding pocket cavity comprising common amino acid residues. Zinc Oxide expressed four hydrogen-bonding interactions with residues ARG 131, ARG 80, TRY 100, and TRY 116, while six hydrogen bonds were observed between cefepime and ARG 131, ARG 80, ARG 43, LUS 21, ASP 111, and LEU 113 residues. Imipenem was shown to establish four hydrogen-bonding interactions with the residues LUS 21, VAL 114, and ASP 111. The final ligand, meropenem, presented three hydrogen bonds with the residues LUS 21, GLY 117, and TRP 299. This indicates that these amino acid residues are necessary for target binding and that they exert activity [22]. A favorable and stable binding position of the compounds within the binding pocket of the target protein is indicated by extremely low energy scores [21,23]. The best binding free energy scores for the four selected compounds show that the highest binding affinity for porin was Zno (−9.1 kcal/mol), followed by cefepime (−7.9 kcal/mol), meropenem (−7.5 kcal/mol), and imipenem (−6.4 kcal/mol). A strong binding property may be induced by the ionic bond, which facilitates enzyme binding. In addition, the aromatic ring, through numerous interaction bonds—including hydrogen, hydrophobic, and ionic bonds—contributes to its great binding affinity with porin protein. Interestingly, with comparable conformations, glide docking scores, and relatively similar binding energies, all four hits fit into the target-binding pocket and interacted with the GLO21 protein similarly; the details of the interactions are presented in Table 2.

To calculate the free energy of the binding of ligands to the target protein, prime molecular mechanics with generalized Born and surface area solvation (MMGBSA) analysis was performed; Table 3, below, illustrates the result. The lowest DG Bind score (most negative) was considered the best DG Bind score [23,24,25]. The MMGBSA binding energies calculated for the tested compounds were ordered as follow: Zinc oxide < cefepime < imipenem < meropenem. The result showed that meropenem had the highest binding energy and, thus, the least binding affinity to GLO21. When compared to additional precision glide docking scores, our results revealed a statistical relationship to experimental binding affinity [20,23,26].

#### 2.2.3. MD Simulation Study

According to the docking data, the four selected compounds and the target protein GLO21 interacted. Thus, MD simulations were carried out on all compounds to confirm the docking analyses and illustrate what occurs in the interacting molecules during a particular time period [23].

##### Stability of Protein–Ligand Complexes

Examining Figure 3, it is apparent that a stable system exists for Zinc Oxide and cefepime, while, in the case of meropenem and imipenem, there are more fluctuations. For the imipenem–GLO21 complex, a very late fluctuation was observed at 98 ns (4.5–11 Å) for the ligand, equilibrating at the end of the trajectory with a slight fluctuation ranging from 9.00 Å To 13.00 Å. On the contrary, the protein showed a stable fluctuation range of 2.0–2.4 Å. Conversely, the Zinc Oxide–GLO21 complex showed a different behavior, with slight fluctuation at the initial phase of the simulation at 5 ns (2.8–6.4 Å) for the ligand, stabilizing within a fluctuation range of 3.5–4.5 Å until the end of the simulation. The protein in the Zinc Oxide–GLO21 system exhibited a stable state of fluctuation for the majority of the simulation time (3.4–6.00 Å). The MRSD value for cefepime in the ligand–protein complex demonstrated major fluctuations, with wide deviations, very early in the simulation (at the first 10 ns (5–13 Å)) and then fluctuated down (10–4.5 Å) during the last 20 ns of the simulation. Alternatively, the RMSD results for the protein reflected minor fluctuations (1.2–10.00 Å) during the 3 ns trajectory, and the protein presented a fluctuation range of 1.8–9.00. The final compound, meropenem, showed many fluctuations for the ligand and protein complex, with the initial one at 5 ns (11.5–20 Å). Overall, Zinc Oxide demonstrated stable complexes with GLO21; however, moderate stability was observed in the cefepime–GLO21 complex, and less stable complexes were formed by meropenem and imipenem. In addition, Figure 4 clarified RMSF graphs for the protein and ligand complexes, reflecting positional differences over time. There was a significant difference in the Zinc Oxide ligand, which presented an amino acid residue index between 50 and 70, increasing the fluctuation of the protein up to 7.0 Å, while the remainder of the ligands showed exclusive fluctuations in various ranges. The overall RMSF values for the protein alone, as well as when it interacted with ligands, were in the range of 3.6 Å to 4.8 Å, with the exception of Zinc Oxide. These results are fairly similar to those obtained by Hendi and others [22].

##### Protein and Ligand Properties from MD Simulation Analysis

The molecular surface area (MolSA), radius of gyration (rGyr), polar surface area (PSA), and solvent-accessible surface area (SASA) values, which reveal information on the behavior of the ligand inside the binding pocket of the GLO21 protein, were calculated to evaluate the ligand’s properties. The RMSD representation is mostly used for protein stability analysis and conformational change prediction in proteins, which leads to the prediction of structural stability [23]. The RMSD values of cefepime and Zinc Oxide primarily fluctuated very early, at 3 ns, and their complex with GLO21 appeared in the ranges from 1.00 Å to 3.00 and 0.2–0.4 Å, respectively. Alternatively, meropenem and imipenem fluctuated at 10 ns and the RMSD values for both were 0.5–2.4 (Figure 5). Although Zinc Oxide displayed the most interesting aspect on this graph, with the best binding pose (less than 2 Å) [20,23]. there was no certainty that the binding to GLO21 was occurring within the candidate protein’s catalytic binding pocket. Therefore, RMSD value was not sufficient to determine the superior binding configuration. Additionally, it was discovered that contact-based analysis of the radius of gyration (rGyr), another parameter, is more reliable and appropriate, as it reflects the protein structure’s compactness [26]. Zinc Oxide and cefepime exhibited slight fluctuation within an acceptable range, and then they steadily reached stability. Alternatively, meropenem and imipenem showed irregular fluctuations, results typical of an unstable protein. The protein was kept compact and folded gradually during the simulation path, based on the steady value of rGyr (Figure 6). On the other hand, cefepime’s greatest MolSA value (400–424 Å^2^) occurred with less fluctuations, while Zinc Oxide and meropenem showed moderate MolSAs (3530–350 Å^2^) with moderate fluctuations. Imipenem demonstrated the lowest MolSA (225–282 Å^2^) with the widest fluctuations. The SASA values of the fourth ligand were assessed, presenting variable results among the candidate ligands. Zinc Oxide was measured as the most constant compound (160–320 Å^2^, with equilibrium at 150 Å^2^). Imipenem had a similar SASA to Zinc Oxide (160–320 Å^2^), but its equilibrium was at 200 Å^2^. Cefepime and meropenem exhibited convergent SASA results (200–450 Å^2^ and 160–400 Å^2^, respectively, with equilibrium at 300 Å^2^ for both). Another important value during the simulation of all tested compounds was PSA, which was measured as 460, 220, and 200 Å^2^ in cefepime, imipenem, and meropenem, respectively. The finding of Zinc oxide’s PSA was also reported by Hendi and others [27]. Interestingly, the Zinc Oxide ligand displayed the best PSA value, confirming the potential of ZnO as an antibacterial drug. This result was supported by the findings of Mugumbate and Overington, who reported that “more polar compounds are a key part of successful future antibacterial discovery” [28].

##### Protein Ligand Contacts and Interacting Bond

Throughout the simulation, interactions between the GLO21 protein and the candidate ligands were observed. The protein–ligand contacts involved hydrophobic bonds, hydrogen bonds, ionic bond, and water bridges. In our results, the most common bond among the four tested compounds was the hydrogen bond, which performs a critical role in protein binding to ligands. An ionic bond was observed mostly with Zinc Oxide. Hendi and others found that only Zinc Oxide presented ionic bond when in contact with SARS-CoV-2 vital proteins [27,29]. For cefepime, the protein–ligand interaction involved hydrogen bonding with 14 residues: SER23, TYR41, ARG80, ARG43, ASP191, GLN258, GLY262, TRP299, TYR301, LYS304, ASN305, ASN307, VAL339, and GIN341 (Figure 6A). For Zinc Oxide, the H-bonds were formed with 18 residues: GLU20, TRP22, SER23, GLY24, GLU25, ASP105, GLU107, TYR340, PHE342, ASP111, MET112, LEU113, VAL114, GLU115, GLY118, ASP119, ASN122, and TYR309 (Figure 6D). Our Zinc oxide–GLO21 complex possessed more residues than those reported by Hendi and others, and the reason behind this variable finding could be due to the protein’s type [20,23]. The third compound (meropenem) constructed the hydrogen bond with 13 residues: NET19, TYR41, ARG43, GLN256, GLN258, ASP260, ARG264, TYR301, LYS304, ASN305, ASN307, and GLN341 (Figure 6C). Imipenem presented hydrogen bonding contact with the largest number of residues, including: LYS21, SER23, GLU25, TYR41, ARG43, ASN65, ARG80, TYR100, TYR104, ASP111, LEU113, VAL114, GLU115, TRP116, GLY117, GLY118, ASP119, ARG131, TRP299, TYR301, ASN307, TYR309, and GLN341 (Figure 6B). Zinc Oxide showed ionic bonding contact with the protein with the following nine residues: GLU20, GLU25, GLU107, ASP105, ASP111, VAL114, GLU115, ASP119, and PHE342. The remainder of the tested compounds had no significant number of ionic bonds. The hydrophobic bond appeared only in the interaction between cefepime, meropenem, and imipenem with ARG264, TYR301, and TYR309, respectively.

Two panels make up the ligand–receptor interaction (histogram) (Figure 6). The panel was divided into top and bottom; each unique contact between the protein and ligand for each trajectory frame is shown in the top panel. The contact numbers changed throughout the trajectory from 0 to 9 (Figure 6). The individual amino acid contributing to the interaction with the ligand was examined in the bottom panel (Figure 7). Each trajectory frame was used to pinpoint the precise amino acid that interacted with the ligand and the stability of the interaction. Some amino acid residues engaged in several precise contacts with the ligand in a specific trajectory framework, represented by the darker orange shade. The results were identical to the histogram data represented in Figure 6. Indeed, an interesting finding was shown in Figure 7D that contributed to the strong evidence of ionic bond role of hydrogen in the interaction between Zinc Oxide and the protein, which reflected robust and stable interactions with GLO21 in the similar six residues previously observed in Figure 6D. Interestingly, our result illustrated that Zinc Oxide was more stable in contact with the target protein than the Zinc Oxide of Hendi and others, which had only one residue interact with Zinc Oxide. A possible explanation for these results may be the ligand’s type and the method of Zinc Oxide synthetization [27,30,31].

A Ramachandran plot, usually used to evaluate the amount of protein alteration after the interaction with the four tested compounds, was the final part of MD simulation analysis. The blue dots scattered in the red and brown area indicate that there was no significant alteration occurred in the GLO21 protein [32]. Overall, all ligands presented similar results, which showed that 89.3% of residues were scattered across the most favored regions; however, 10.3% of residues were scattered across permitted regions and 1% of residues were scattered among forbidden regions. These results indicated that no significant alteration occurred and there was little damage to the target protein (Figure 8).

### 2.3. Antibacterial Activity

Bioinformatics was employed as a preliminary screening study to assess the antibacterial activity of produced zinc oxide nanoparticles and β-lactam antibiotics. The computational simulation of the structure of the selected ligands and protein found that Zinc Oxide is the most potent compound, based on MD results and the interaction energy generated from this reaction. We used zone of inhibition, MIC, and MBC to test the four compounds against KPC and *Klebsiella pneumoniae* (ATCC 700603) to confirm the findings of the MD study. Then, we studied the alteration of the bacterial surface by SEM.

#### Zone of Inhibition of ZnO-NPs and the Selected Antibiotics

As shown in Table 4 and Figure 9A–C, ZnO-NPs displayed the most significant zone of inhibition among the tested compounds (26 mm with KP and 24 mm with KPC). Cefepime showed 20 mm with KP, and no effect appeared against KPC. Additionally, neither meropenem nor imipenem exhibited any activity against KP and KPC. This finding was also reported by many researchers who tested ZnO-NPs against carbapenem-resistant *Klebsiella pneumoniae* [6,11,33]. The MIC of an antibacterial agent is measured in mg/L (g/mL), the lowest concentration at which the test strain of an organism cannot grow in any way that can be observed [34]. MBC is the lowest concentration of antibiotic, eliminating at least 99.9% of organisms [34]. ZnO-NPs’ MIC presented the lowest MIC and MBC when compared with the tested antibiotics: 0.2 mg/mL, 0.5 mg/mL MIC and MBC, respectively, against KP and KPC; >64 mg/mL MIC and MBC for cefepime against KPC; >16 mg/mL for meropenem’s MIC and MBC against both tested bacteria; and >4 mg/mL MIC and MBC for imipenem against KPC and KP (Table 4). The MIC and MBC results for ZnO-NPs observed in this investigation are far below those observed by our previous study [6,11]. The final in vitro test was the study of bacterial surface alteration before and after treatment by ZnO-NPs and antibiotics using SEM (Figure 10A–C). The images below demonstrate that the effect of ZnO-NPs cause changes in cell size and shape, which resulted in the loss of membrane integrity and cell death (Figure 10C). No significant changes are shown in Figure 10B, which presents the SEM image of KPC treated by imipenem. In summary, there were reports of zinc oxide nanoparticles acting as an antimicrobial agent. Furthermore, a safe and perfect medicine delivery platform would be made of non-toxic, environmentally benign green materials [17]. In agreement with this, in vitro and in silico research revealed a strong inhibitory effect for Zinc Oxide and a weak antibacterial activity for the antibiotics cefepime, meropenem, and imipenem against the KPC GLO21 protein. As a result, the main conclusions of the present study highlight the potential of the nanoparticles ZnO as a stable therapeutic agent against the porin proteins of KPC. Further studies are needed to establish the effectiveness of ZnO-NPs with animal models as antimicrobial agents.

## 3. Methodology

### 3.1. Synthesis of Zinc Oxide Nanoparticles (ZnO-NPs) and Characterization

Based on a method created by Elsayim and others, zinc oxide nanoparticles were synthesized by mixing 20 grams of *Acacia nilotica* fruits with 150 mL of distilled water (DW), boiling the solution 80 °C for 2 h, and then filtered the aqueous mixture. Subsequently, we added 5 gm of zinc nitrate obtained from Sigma-Aldrich Co. (Budapest, Hungary) to 100 mL of the aqueous extract of *A. nilotica* and boiled the mixture until the color changed to a deep brownish yellow paste. The zinc nitrate and aqueous extract mixture was kept. at 60 °C for 24 h, resulting in a dark brown spongy paste. Finally, the spongy paste was formed and packed before being transferred to a muffle furnace at 400 °C for 2 h to obtain white powder of pure ZnO-NPs [6]. After ZnO-NPs formed, we tested their formation by using UV-Vis spectroscopy (UV-1800; Shimadzu UV Spectrophotometer, Kayoto, Japan). Size and stability of ZnO-NPs were measured by dynamic light scattering measurement, performed with a Malvern Zetasizer Nano series compact scattering spectrometer (Malvern Instruments Ltd., Malvern, UK). The shape, structure, and distribution of the synthesized nanoparticles were examined by SEM (JEOL model, JSM-761OF, Tokyo, Japan). Fourier-transform infrared spectroscopy (FTIR), in the range of 400–4000 cm^−1^ (Parkin Elmer, Spectrum BX, Waltham, UK) was used to study the different functional groups present in the aqueous plant extract and ZnO-NPs.

### 3.2. Protein Targets and Ligands

Receptor: porin protein (Uniprot ID: A0A871B143 A0A871B143_KLEPN) ligands: bioactive compounds in zinc oxide, meropenem, imipenem, and cefepime.

### 3.3. Molecular Docking Study

Molecular docking is used to forecast the interactions that resulted in the binding protein and the four phytochemicals (ligands). The biological targets for the docking investigation were taken from the three-dimensional structure of the porin protein.

### 3.4. QSAR Studies

In order to predict the reactivity and characteristics of the chosen compounds, QSAR research was used. The Hyper Chem Professional 8.0.3 application was used to perform computational calculation (Hypercube, Gainesville, FL, USA). Energy reduction was accomplished using a Fletcher–Reeves conjugate gradient algorithm approach, and the compounds with a strong docking score were first optimized using the (MM+) force field, with semi-empirical PM3 methods [20,35].

### 3.5. Molecular Dynamics Simulation

The three selected antibiotics were used as ligands for additional MD simulation study to compare with ZnO-NPs in the treatment of KPC. To learn more about the stability of the protein–ligand complexes, these compounds underwent a 100 ns molecular dynamics simulation. Studies on molecular dynamics were carried out using ligand docking module of maestro 12.3.The target protein’s active binding site was created using the “Receptor Grid Generation” module before docking. The partial charge cut-off and van der Waals radius scaling factor were set at 1.0 and 0.25, respectively, during this process. The remaining parameters were left at their default values. The four phytocompounds were molecularly docked to the active porin-binding site using the “Extra Precision” (XP) model. It was possible to determine the molecular interaction behavior and binding affinity. With the use of Schrodinger’s Desmond module, MD simulation investigations were conducted. Before beginning the MD simulation, the system underwent an equilibration phase until it achieved a stationary state. For a duration of 100 ns, the MD simulation was run at a temperature and atmospheric pressure of 310 K and 1.013 bar, respectively. A simulation interaction diagram was used to thoroughly assess the MD simulation’s outcomes. The protein–ligand interaction diagram, the root-mean-square deviation of the protein–phytocompound complex, the root-mean-square fluctuation of the protein, the interacting amino acid residues with the ligand in each trajectory frame, and the trajectory of various ligand properties were all examined. Ramachandran plot analysis was used to further evaluate the protein’s overall stability [22].

### 3.6. Antibacterial Activity

The KPC samples were obtained from Prince Mohammed bin Abdul Aziz Hospital, Al Madinah. The samples were identified in the hospital’s Microbiology Department utilizing the VITEK 2 system, version 08.01. The antibacterial activity of ZnO-NPs was evaluated by the agar well diffusion method [35]. Imipemun, meropenem, and cefepime disks (10 µ concentration) were used as controls. By using a broth macrodilution process, the MIC and the MBC were established in accordance with the Clinical and Laboratory Standards Institute (CLSI) protocol [36].

## 4. Conclusions

The aim of the present research was to examine the activity of ZnO-NPs, meropenem, imipenem, and cefepime against KPC in silico and in vitro. The findings clearly indicate that the ecofriendly synthesized nanoparticles outperform antibiotics for treating multidrug-resistant bacteria. The research has also shown that molecular docking and simulation are effective methods for assessing a compound’s capacity to interact with the target pathogen, making MD an important component in the drug discovery process. Overall, this study strengthens the idea that before initiating in vitro and in vivo investigations of any compound and protein interactions, it is necessary to use simulation and MD software to reduce experiment costs and time loss. This study provides the first MD and simulation of ZnO-NPs against KPC. However, these results may not be applicable to all types of synthesized nanoparticles. Although the current study is based on a small sample of tested bacteria, the findings suggest that ZnO-NPs possess the potent characteristics of a good antibacterial drug. It would be interesting to assess the effects of ZnO-NPs in vivo, particularly as a systemic drug.

## Figures and Tables

**Figure 1 molecules-28-02510-f001:**
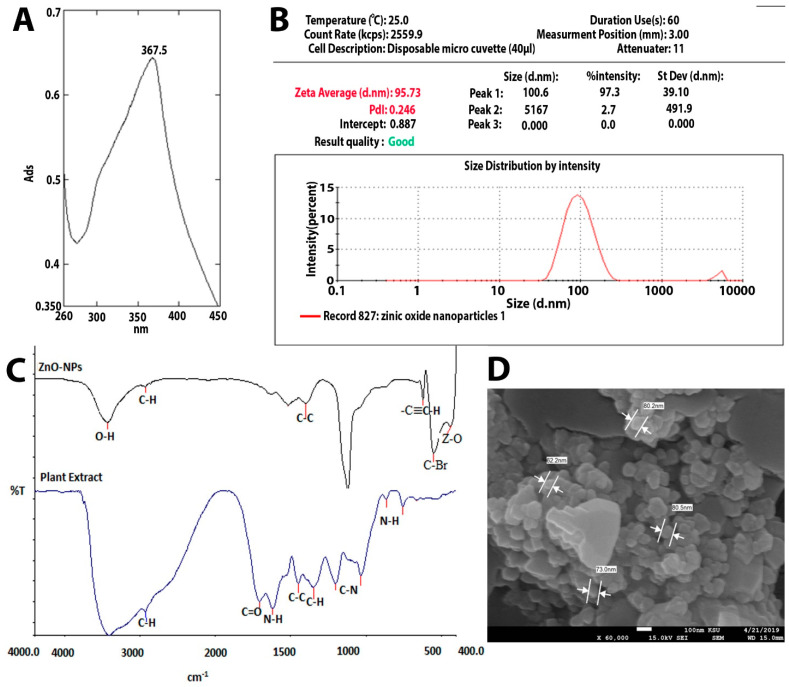
(**A**) UV-visible spectrum of ZnO-NPs. (**B**) Zetasizer of ZnO-NPs. (**C**) FTIR of ZnO-NPs. (**D**) SEM of ZnO-NPs and (**E**) zeta potential of ZnO-NPs.

**Figure 2 molecules-28-02510-f002:**
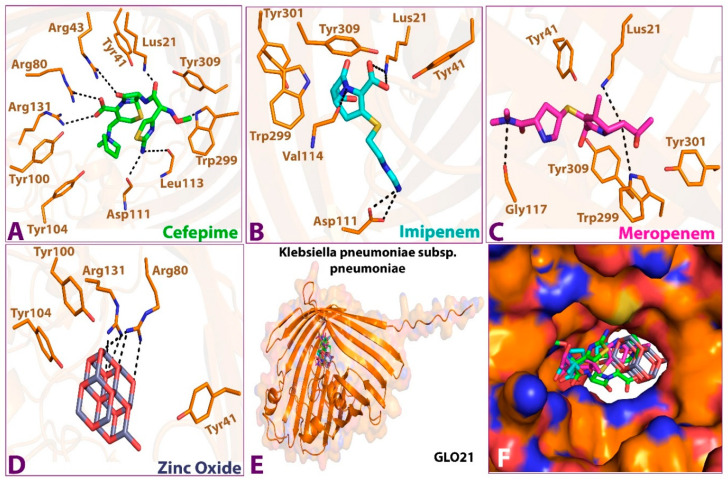
Interactions between the four compounds: (**A**,**A’**) Cefepime, (**B**,**B’**) Imipenem, (**C**,**C’**) Meropenem, and (**D**,**D’**) Zinc Oxide, and the target protein GLO21. (**E**) interaction shape showing all compounds and (**F**) binding cavity of the protein showing all compounds, presented in 3D (**A**–**E**) and 2D (**A’**–**D’**) shapes.

**Figure 3 molecules-28-02510-f003:**
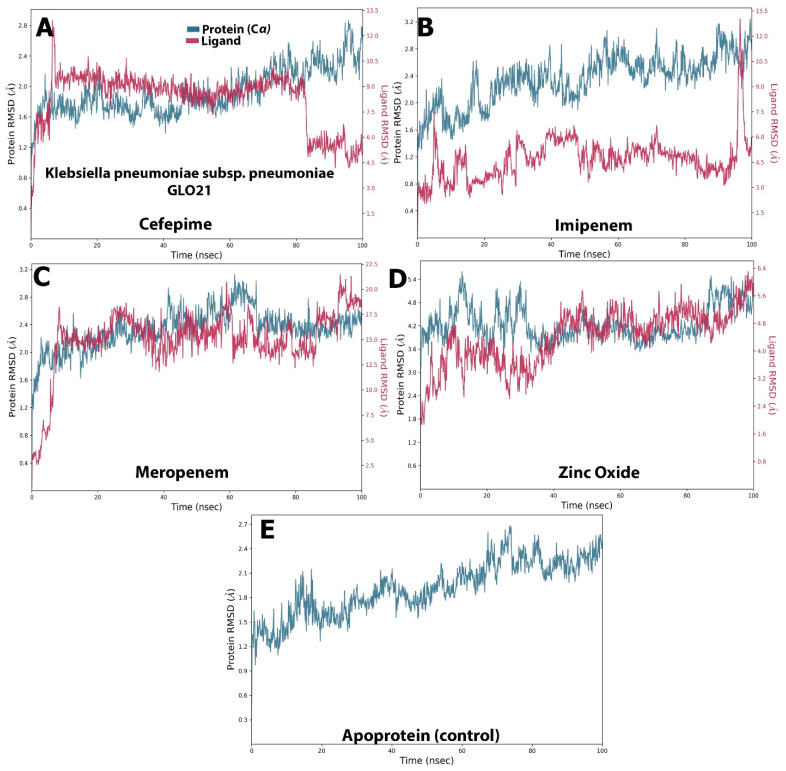
The selected compounds’ protein–ligand RMSD over 100 ns. (**A**) Cefepime, (**B**) imipenem, (**C**) meropenem, (**D**) zinc oxide, (**E**) apo-form as a control.

**Figure 4 molecules-28-02510-f004:**
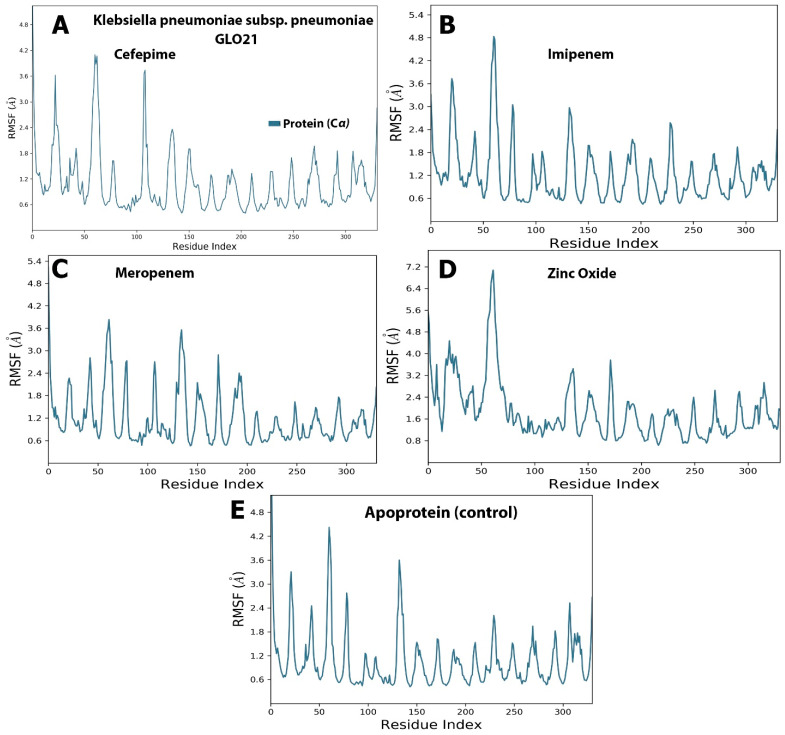
The selected compounds’ protein–ligand RMSF over 100 ns. (**A**) Cefepime, (**B**) imipenem, (**C**) meropenem, (**D**) zinc oxide, (**E**) apo-form as a control.

**Figure 5 molecules-28-02510-f005:**
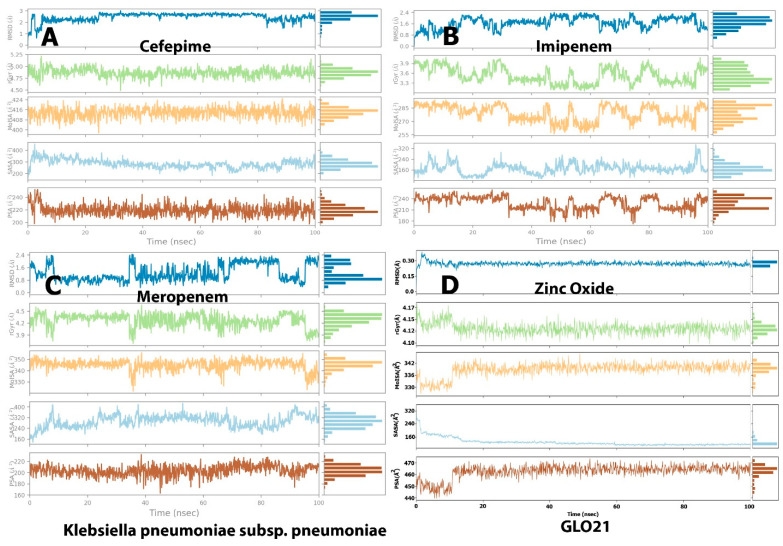
Post-molecular dynamic simulation analysis of GLO21 and ligand properties. Radius of gyration (rGyr); molecular surface area (MolSA); solvent-accessible surface area (SASA); and polar surface area (PSA). (**A**) Cefapime, (**B**) Imipenem, (**C**) Meropenem and (**D**) Zinc Oxide.

**Figure 6 molecules-28-02510-f006:**
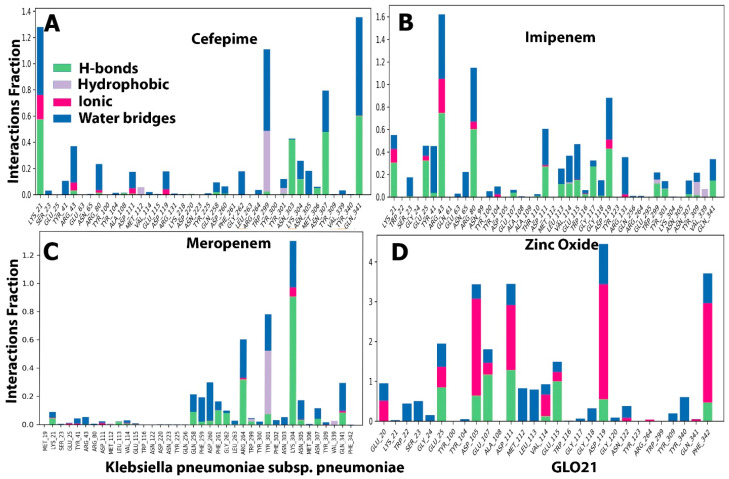
The histogram of protein–ligand (GLO21, four selected compounds, (**A**–**D**)) contact throughout the trajectory.

**Figure 7 molecules-28-02510-f007:**
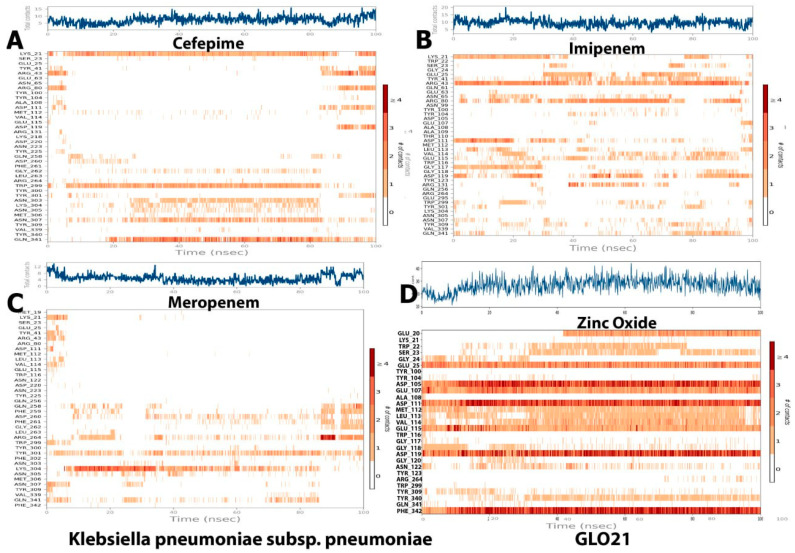
Ligand–protein contact plot for compounds/GLO21 complexes during simulation trajectory. (**A**) Cefapime, (**B**) Imipenem, (**C**) Meropenem and (**D**) Zinc Oxide.

**Figure 8 molecules-28-02510-f008:**
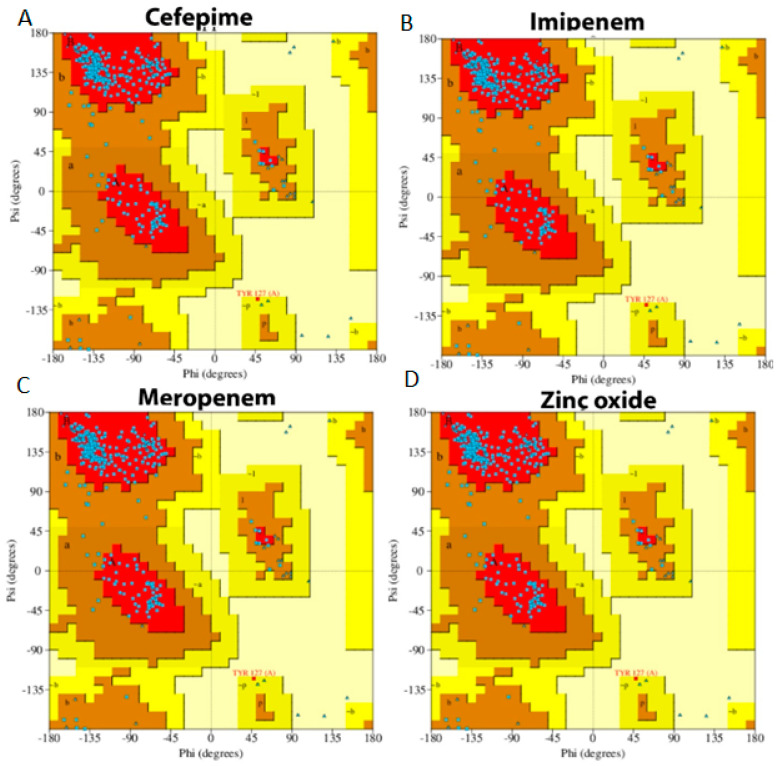
Ramachandran plot of protein interacting with the four selected compounds. (**A**) Cefepime, (**B**) imipenem, (**C**) meropenem, (**D**) zinc oxide.

**Figure 9 molecules-28-02510-f009:**
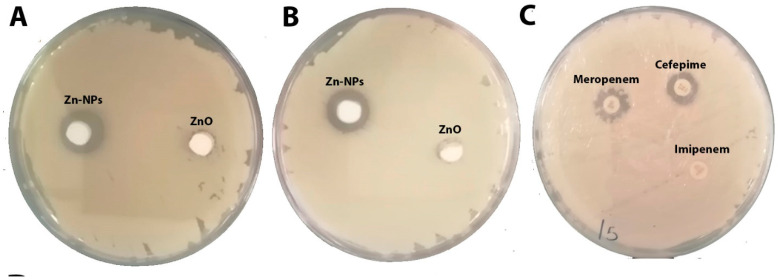
(**A**) Zone of inhibition of ZnO-NPs against *Klebsiella pneumoniae* (ATCC 700603) at concentrations of 7.5 mg/mL. (**B**) Zone of inhibition of ZnO-NPs against KPC at concentrations of 7.5 mg/mL. (**C**) Zone of inhibition of cefepime, meropenem and imipenem discs against KPC.

**Figure 10 molecules-28-02510-f010:**
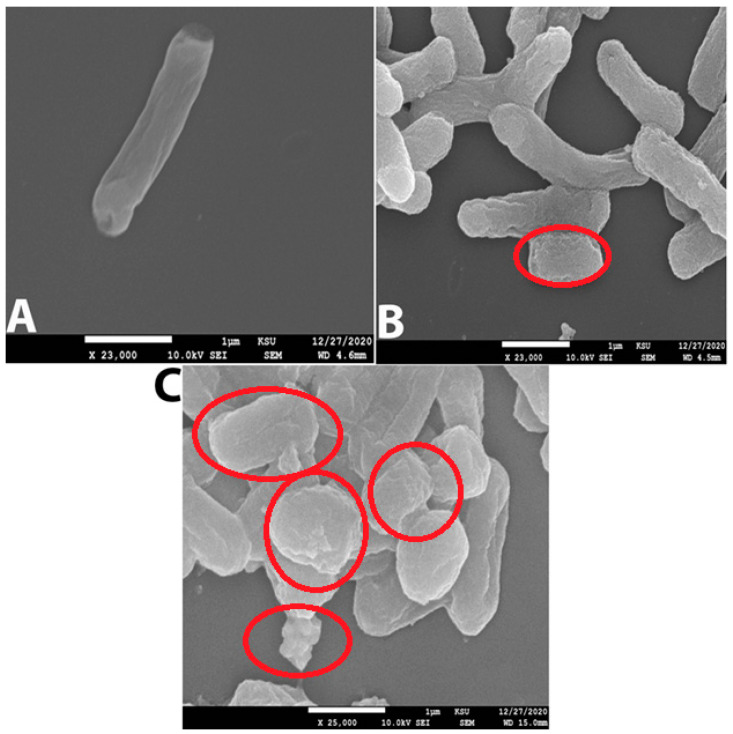
(**A**) Normal cell of KPC without treatment (Control). (**B**) KPC treated with imipenem 500 mg/mL; the red circle indicates the cell shrinking. (**C**) KPC treated with ZnO-NPs 0.2 mg/mL; red circles show more cell alterations, such as shrinking, loss of membrane integrity, and cell death.

**Table 1 molecules-28-02510-t001:** QSAR data for the tested compounds.

Function	Cefepime	Imipenem	Meropenem	Zinc Oxide
Surface area (Approx) (Å^2^)	579.69	474.86	571.50	605.19
Surface area (Grid) (Å^2^)	703.60	544.04	620.34	465.81
Volume (Å^3^)	1226.74	871.98	1068.58	752.79
Hydration energy (Kcal/mole)	−15.20	−19.58	−11.69	−101.84
Log P	6.79	2.87	5.68	0.50
Refractivity (Å^3^)	82.56	54.10	79.95	5.52
Polarizability (Å^3^)	45.93	29.14	38.04	2.97
Mass (amu)	480.56	299.34	383.46	1058.81
Total energy (kcal/mol)	88.3957	53.6913	67.7658	187.262
Dipole moment (Debye)	0.3984	2.234	0.7063	1.115
RMS gradient (kcal/Å mol)	0.09671	0.09816	0.09849	0.08826

**Table 2 molecules-28-02510-t002:** Interactions and binding energies of ligands with GLO21 from Klebsiella pneumoniae subsp. pneumoniae.

Sl. No.	Ligand	Receptor-Chain (A)	Interaction	Distance	E (kcal/mol)	Binding Energy ∆G (Autodock Vina)
Cefepime	N4 19	OD1 ASN 307	(A) H-donor	2.88	−1.9	−7.9
	N5 27	OE1 GLU 115	(A) H-donor	2.60	−12.7	
	S2 30	SD MET 112	(A) H-donor	3.49	−0.5	
	N6 33	O MET 112	(A) H-donor	2.94	−1.0	
	N6 33	SD MET 112	(A) H-donor	3.34	−4.5	
	N6 33	O LEU 113	(A) H-donor	2.75	−5.8	
	C2 52	OD2 ASP 111	(A) H-donor	3.33	−0.8	
	C2 52	O GLY 117	(A) H-donor	3.17	−0.7	
	C6 55	O GLU 115	(A) H-donor	3.04	−1.0	
	O1 2	NH1 ARG 43	(A) H-acceptor	2.77	−3.7	
	O2 37	NH1 ARG 80	(A) H-acceptor	2.75	−9.1	
	O2 37	NH2 ARG 80	(A) H-acceptor	2.79	−4.9	
	O3 38	NH2 ARG 43	(A) H-acceptor	2.68	−11.6	
	N4 19	OE1 GLU 115	(A) ionic	3.24	−3.1	
	N4 19	OE2 GLU 115	(A) ionic	3.32	−2.7	
	N5 27	OE1 GLU 115	(A) ionic	2.60	−7.8	
	O2 37	NH1 ARG 80	(A) ionic	2.75	−6.4	
	O2 37	NH2 ARG 80	(A) ionic	2.79	−6.1	
	O2 37	NE ARG 131	(A) ionic	3.35	−2.5	
	O2 37	NH1 ARG 131	(A) ionic	3.44	−2.1	
	O2 37	NH2 ARG 131	(A) ionic	3.17	−3.4	
	O3 38	NH1 ARG 43	(A) ionic	3.10	−3.8	
	O3 38	NH2 ARG 43	(A) ionic	2.68	−7.0	
	O3 38	NH1 ARG 80	(A) ionic	2.94	−4.9	
	O3 38	NH2 ARG 80	(A) ionic	3.12	−3.7	
Imipenem	S1 13	OD2 ASP 111	(A) H-donor	3.65	−0.7	−6.4
	N2 20	OE1 GLU 115	(A) H-donor	2.80	−11.7	
	C12 21	O GLY 117	(A) H-donor	3.03	−2.0	
	N3 24	O GLU 115	(A) H-donor	2.76	−3.9	
	N3 24	OD1 ASP 119	(A) H-donor	2.81	−8.9	
	O3 29	OE2 GLU 115	(A) H-donor	2.48	−0.4	
	O4 28	NH1 ARG 43	(A) Hacceptor	2.87	−3.7	
	O4 28	NH2 ARG 43	(A) Hacceptor	3.35	−0.7	
	O3 29	NH2 ARG 43	(A) Hacceptor	2.85	−1.7	
	O1 37	NE1 TRP 299	(A) Hacceptor	2.89	−1.2	
	N2 20	OE1 GLU 115	(A) ionic	2.80	−6.0	
	N3 24	OD1 ASP 119	(A) ionic	2.81	−5.9	
Meropenem	C13 22	OD2 ASP 111	(A) H-donor	3.04	−1.9	−7.5
	N2 24	OD1 ASP 111	(A) H-donor	2.65	−9.3	
	N2 24	O GLY 117	(A) H-donor	2.71	−6.2	
	O3 43	OE1 GLU 115	(A) H-donor	2.43	6.3	
	O5 31	NH1 ARG 131	(A) H-acceptor	2.88	−2.2	
	O5 31	NH2 ARG 131	(A) H-acceptor	2.94	−3.7	
	O4 42	NH1 ARG 43	(A) H-acceptor	2.82	−3.2	
	O4 42	NH1 ARG 80	(A) H-acceptor	2.88	−0.9	
	O4 42	NH2 ARG 80	(A) H-acceptor	2.88	−1.8	
	O2 51	NE1 TRP 299	(A) H-acceptor	2.82	−1.7	
	N2 24	OD1 ASP 111	(A) ionic	2.65	−7.3	
	N2 24	OD2 ASP 111	(A) ionic	3.58	−1.6	
ZnO	Ionic and hydrophobic interaction					−9.1

**Table 3 molecules-28-02510-t003:** Binding energies (MMGBSA) of porin protein and the selected compounds.

Compound	MMGBSA dG Bind(NS)	MMGBSA dG Bind(NS) Coulomb	MMGBSA dG Bind Covalent	MMGBSA dG Bind(NS) Hbond	MMGBSA dG Bind(NS) Lipo	MMGBSA dG Bind(NS) Solv GB	MMGBSA dG Bind(NS) vdW
Zinc oxide	1071.258	−7842.09	−107.921	−168.326	−612.175	8946.66	74.61135
Meropenem	−33.7034	22.20441	0.350149	−3.27054	−5.90737	−23.6331	−23.4469
Imipenem	−24.3949	9.574381	0.060334	−2.54655	−9.49918	3.736621	−25.7205
Cefepime	−27.8693	−28.3957	1.987949	−1.02687	−8.06542	45.62215	−37.9914

**Table 4 molecules-28-02510-t004:** Zone of inhibition (ZI), MIC, and MBC of ZnO-NPs and the selected antibiotics against Klebsiella pneumoniae (ATCC 700603) and tested bacteria.

Bacterial Code	ZnO-NPs			Cefepime			Meropenem			Imipenem		
Test	ZI mm	MIC mg\mL	MBC mg\mL	ZI mm	MIC mg\mL	MBC mg\mL	ZI mm	MIC mg\mL	MBC mg\mL	ZI mm	MIC mg\mL	MBC mg\mL
KP-ATCC	26	0.2	0.5	20	0.9	0.2	I	<0.25	0.5	R	<1	1
KPC	24	0.2	0.9	R	>64	>64	R	>16	>16	R	>4	>4

## Data Availability

The data presented in this study are available on request from the corresponding author.

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
