# Peer review of "Molecular Dynamic Analysis of Carbapenem-Resistant Klebsiella pneumonia’s Porin Proteins with Beta Lactam Antibiotics and Zinc Oxide Nanoparticles"

_molecules, 2023, doi:10.3390/molecules28062510_

Round 1
Reviewer 1 Report
1. The size distribution of ZnO NPs is 95.73; however, as shown in Figure 1D, is very heterogeneous, and the authors only label small size particles.
2. Figure 2, A~D lables were duplicated
3. In Figure 3, the lables is missing
4. In Figure 10, bacterial morphology chages need to be marked in detail.
5. the unit "mg\ml" should be "mg/ml"; also the "Kcal\mol"
Reviewer 2 Report
This paper “Molecular dynamic analysis of carbapenem resistant Klebsiella pneumonia’s porins protein with Beta lactam antibiotics and Zinc Oxide nanoparticles” is very well-presented, structured and organized.
The study used molecular dynamics to examine how ZnO-NPs interact with the porin protein, a target of β-lactam antibiotics, and then tested this interaction in vitro by determining the zone of inhibition, minimum inhibitory concentration, and minimum bactericidal concentration, as well as the alteration of KPC's cell surface.
Even if the nanoparticles produced was characterized by UV- vis spectroscopy, zetasizer, Fourier transform infrared spectroscopy (FTIR) and scanning electron microscopy (SEM). However, this study suffers from deficiencies in the determination of the surface properties of ZnO nanoparticles. I suggest to authors to more characterize these nanoparticles, especially, their specific interactions, dispersive surface properties, specific surface area, zeta potential and their Lewis- acid base properties. These determinations can help readers to understand more the useful role of Zinc Oxide nanoparticles and their activity of ZnO-NPs relatively to antibiotics for treating of multi-drug resistant bacteria.
Round 2
Reviewer 2 Report
Accepted